

# Impact of body weight gain on hepatic metabolism and hepatic inflammatory cytokines in comparison of Shetland pony geldings and Warmblood horse geldings

Carola Schedlbauer[1], Dominique Blaue[1], Martin Gericke[2], Matthias Blüher[3], Janine Starzonek[1], Claudia Gittel[4], Walter Brehm[4] and Ingrid Vervuert[1]

[1] Leipzig University, Institute of Animal Nutrition, Nutrition Diseases and Dietetics, Leipzig, Saxony, Germany
[2] Leipzig University, Institute of Anatomy, Leipzig, Saxony, Germany
[3] Leipzig University, Department of Medicine, Leipzig, Saxony, Germany
[4] Leipzig University, Department for Horses, Leipzig, Saxony, Germany

Corresponding author
Ingrid Vervuert,
ingrid.vervuert@vetmed.
uni-leipzig.de

## ABSTRACT

**Background:** Non-alcoholic fatty liver disease is known as determining part of human obesity. The impact of body weight (BW) gain on liver metabolism has not been extensively investigated yet.

**Objectives:** To investigate hepatic alterations caused by increasing BW in ponies and horses.

**Animals:** A total of 19 non-obese equines (10 Shetland ponies, geldings; nine Warmblood horses, geldings).

**Methods:** Animals received 200% of their metabolizable maintenance energy requirements for 2 years. Serum alkaline phosphatase, glutamate dehydrogenase (GLDH), aspartate aminotransferase (AST), and gamma-glutamyl transferase activities and bile acids were analyzed several times during 2 years of hypercaloric diet. Hepatic lipid content and hepatic levels of the interleukin (IL)-6, tumor necrosis factor $\alpha$ (TNF$\alpha$), cluster of differentiation (CD) 68, IL-1$\beta$, lipoprotein lipase (LPL), fatty acid-binding protein 1, chemerin and nuclear factor-$\kappa$B mRNAs were assessed at the start of the study and after 1 and 2 years of excess energy intake.

**Results:** The mean ($\pm$SD) BW gain recorded during 2 years of excess energy intake was 29.9 $\pm$ 19.4% for ponies and 17 $\pm$ 6.74% for horses. The hepatic lipid content was not profoundly affected by increasing BW. Levels of the IL-6, TNF$\alpha$, CD68 and IL-1$\beta$ mRNAs did not change during BW gain. Levels of the chemerin mRNA increased significantly in both breeds (ponies: $P = 0.02$; horses: $P = 0.02$) in response to BW gain. Significant differences in serum GLDH and AST activities, serum bile acid concentrations and hepatic levels of the LPL mRNA were observed between ponies and horses at the end of the study.

**Conclusions:** Chemerin might represent an interesting marker for future equine obesity research. Interestingly, steatosis caused by increasing BW may occur later in the development of obesity in equines than in humans. Additionally, the hepatic metabolism exhibits differences between ponies and horses, which may explain in part the greater susceptibility of ponies to obesity-associated metabolic dysregulations.

## INTRODUCTION

Obesity is an increasing problem in humans and companion animals, such as horses. Metabolic syndrome (metS) in humans is characterized by the accumulation of different symptoms, namely, obesity, increased circulating triglycerides, reduced high density lipoprotein–cholesterol concentrations, increased blood pressure and increased fasting glucose levels (*Alberti, Zimmet & Shaw, 2006*). Equines develop a similar but not identical symptom complex termed equine metabolic syndrome (EMS), which is defined by obesity, insulin dysregulation and a predisposition toward laminitis (*Frank et al., 2010*). EMS is furthermore linked to dyslipidemia, hyperleptinemia, arterial hypertension and low-grade inflammation (*Frank et al., 2010*). Susceptibility to EMS seems to be higher in pony breeds than in most horse breeds (*Johnson et al., 2010*).

The association between metS and the liver has been studied extensively in humans, as the livers of individuals suffering from metS exhibit frequently a form of steatosis termed nonalcoholic fatty liver disease (NAFLD). Some authors consider NAFLD to be the hepatic manifestation of metS (*Cortez-Pinto et al., 1999*). On the other hand, NAFLD also appears to be a precursor of metS and type 2 diabetes. Therefore, NAFLD seems to be a risk factor for the development of metS (*Lonardo et al., 2015*). Among other parameters, NAFLD is characterized by increased serum liver enzyme activities, such as aminotransferases (*Sookoian et al., 2016*). Hence, elevated serum liver enzyme activities emerged as potential biomarkers of an increased risk for developing metS and its related complications (*Devers et al., 2008*; *Zhang et al., 2015*). To the best of the authors' knowledge, the interrelations of equine obesity and the liver have not been studied yet. One study reported serum gamma-glutamyl transferase (GGT) and aspartate aminotransferase (AST) activities that exceeded the reference ranges in obese horses with a history of laminitis (*Chameroy et al., 2011*). The authors suggested that hepatic lipidosis caused the changes in liver metabolism.

In addition, high circulating levels of proinflammatory cytokines (e.g., tumor necrosis factor α (TNFα), interleukin-1 (IL-1), and interleukin-6 (IL-6)), markers of lipid metabolism (e.g., fatty acid-binding protein 1 (FABP1)) and adipokines (e.g., leptin, chemerin) have been reported in equine and human obesity (*Vozarova et al., 2001*; *Bozaoglu et al., 2007*; *Vick et al., 2007*; *Shi et al., 2012*; *Qu, Deng & Hu, 2013*). Adipose tissue is thought to be the main site of production of these adipokines (*Hotamisligil, Shargill & Spiegelman, 1993*; *Arner, 2005*; *Blüher, 2012*) although resident liver macrophages (Kupffer cells) may also play a prominent role (*Baffy, 2009*). Likewise, chemerin, another potent marker of inflammation, seems to be synthesized in the liver rather than in the visceral adipose tissue of human patients with liver cirrhosis (*Weigert et al., 2010*). However, researchers have not determined whether hepatic levels of the above-mentioned mediators might be affected by equine obesity.

The aim of the present study was to investigate changes in serum liver enzyme activities, serum bile acids (BA), liver lipid content and hepatic mRNA levels of several markers
of inflammation and lipid metabolism in the course of increasing body weight (BW) in equines. The comparison of Shetland ponies and Warmblood horses should elucidate the underlying reasons for the higher predisposition of pony breeds to metabolic derangements. We hypothesized that equine obesity is associated with hepatic alterations. Furthermore, we expected that liver metabolism exhibits different responses between ponies and horses during long-term excess energy intake.

## MATERIALS AND METHODS

### Animals

Ten Shetland ponies (geldings; mean age 6 ± 3 years, *Equus caballus*) and nine Warmblood horses (geldings; mean age 10 ± 3 years, *E. caballus*) owned by the Institute of Animal Nutrition, Nutrition Diseases and Dietetics of the Leipzig University were included in the study. All animals were supposed to be adult, therefore we included equines older than 3 years and younger than 15 years. We decided to use only geldings to exclude the influence of gender related differences such as the sexual cycle in mares. Prior to the study, pituitary pars intermedia dysfunction was excluded by measuring adrenocorticotrophic hormone (ACTH) levels after 8 h of fasting. An experienced clinician (CG) confirmed the absence of clinical or radiological signs of previous or acute laminitis of the front feet in all animals. The animals were bedded on straw in individual box stalls and were turned out onto a dry lot for approximately 5 h a day. The animals were adapted to the experimental conditions for at least 2 weeks. The Ethics Committee for Animal Rights Protection of the Leipzig District Government (No. TVV 32/15) approved the project in accordance with German legislation for animal rights and welfare. Animals were cared for according to the guidelines for the accommodation and care of animals used for experimental and other scientific purposes (2007/526/EC).

### Study design

The study was conducted from October 2015 until December 2017. Ponies and horses initially received meadow hay and a commercial mineral supplement to meet or exceed their energy and nutrient requirements during maintenance according to the guidelines of the Society of Nutrition Physiology (GfE 2014) (*Flachowsky et al., 2014*). The basal state of the animals was assessed in October 2015 (t0) by examination of serum liver enzyme activities, serum BA, serum amyloid A (SAA), plasma glucose, serum insulin, serum triglycerides (TG), serum non-esterified fatty acids (NEFA) concentrations, by conduction of a combined glucose insulin test (CGIT) (*Eiler et al., 2005*) and by liver tissue sampling. Following the collection of these initial data, the animals underwent a feeding period by receiving 200% of their metabolizable energy (ME) maintenance requirements according to the GfE (2014) under conditions of gradual adaptation (*Flachowsky et al., 2014*). A total of 60% of the energy intake was supplied by hay, and 40% was provided by a concentrate (Pavo Pferdenahrung GmbH, Vechta Langförden, Germany). For dietary intake of nutrients and intake of ME see Table 1. BW, body condition score (BCS) and cresty neck score (CNS) were monitored weekly. Energy intake was adapted monthly to the current BW. During the feeding period further data were collected in July 2016 (t1),

Table 1 Estimated dietary intake per equine on a daily basis and calculated dietary composition during the whole feeding period (data are presented as mean ± SD).

| Variable | | Ponies | Horses |
|---|---|---|---|
| Feed intake (kg DM/100 kg BW) | Meadow hay | 1.95 ± 0.16 | 1.53 ± 0.13 |
| | Concentrate | 0.54 ± 0.08 | 0.48 ± 0.07 |
| Nutrient intake (% of dry matter intake) | Crude fat | 4.42 ± 0.42 | 4.70 ± 0.41 |
| | Crude protein | 9.07 ± 1.85 | 9.20 ± 1.80 |
| | Crude fiber | 29.1 ± 2.62 | 28.5 ± 2.50 |
| | Starch | 7.45 ± 0.11 | 8.20 ± 0.12 |
| | Sugar | 9.91 ± 1.07 | 9.45 ± 0.77 |
| ME intake (% of maintenance requirements) | | 199 ± 0.20 | 185 ± 13.1 |

October 2016 (t2), April 2017 (t3), July 2017 (t4) and December 2017 (t5). At all time points blood samples were obtained for assessments of serum liver enzyme activities and serum BA. In addition to t0, SAA, plasma glucose, serum insulin, serum TG and serum NEFA concentrations were analyzed at t2 and t5. Additionally, at t2 and t5 the insulin sensitivity was assessed by performing a CGIT (*Eiler et al., 2005*) and a lipopolysaccharide (LPS) challenge was conducted followed by liver tissue sampling 15 h later.

## Blood sampling

Blood samples for assessment of serum insulin, SAA, serum NEFA, serum TG and plasma glucose concentrations were collected at t0, t2 and t5. Blood samples for analysis of serum liver enzyme activities and serum BA concentrations were obtained at t0, t1, t2, t3, t4 and t5. After 8 h of fasting, a 14-gauge-catheter (Milacath; Mila International, Florence, KY, USA) was aseptically placed into the jugular vein of ponies and horses. Blood samples were collected in tubes containing coagulation activator (Monovette; Sarstedt AG, Nuembrecht, Germany) and centrifuged at $865 \times g$ for 10 min after 30 min of clotting time for assessments of serum insulin levels, liver enzyme activities, BA, SAA, NEFA and TG concentrations. Blood samples for plasma glucose assessments were collected in tubes containing sodium fluoride (S-Monovette; Sarstedt AG, Nuembrecht, Germany) and immediately centrifuged at $865 \times g$ for 10 min. Serum and plasma samples were gradually frozen from −20 to −80 °C and stored at −80 °C until analysis.

## CGIT

The CGIT was conducted at t0, t2 and t5 and consisted of rapid IV administration of 150 mg/kg BW of glucose (40% anhydrous glucose; WDT, Garbsen, Germany) and 0.1 U/kg BW of insulin (Humulin R; Lilly USA, Indianapolis, IN, USA) mixed with three mL of 0.9% saline as adapted from *Eiler et al. (2005)*. Blood samples were collected before and 1, 5, 15, 25, 35, 45, 60, 75, 90, 105, 120, 135 and 150 min after insulin injection in tubes containing sodium fluoride (S-Monovette; Sarstedt AG, Nuembrecht, Germany) or a coagulation activator (Monovette; Sarstedt AG, Nuembrecht, Germany). Tubes containing a coagulation activator were centrifuged after a clotting time of 30 min, whereas tubes containing sodium fluoride were immediately centrifuged at

$865 \times g$ for 10 min. Plasma and serum samples were aliquoted and gradually frozen from −20 to −80 °C.

## LPS challenge

The LPS challenge was performed 3–5 days after initial blood sampling. A 14-gauge-indwelling catheter (Milacath; Mila International, Florence, KY, USA) was aseptically inserted into the jugular vein. LPS (*Escherichia coli* 055:B5, 1,000 ng/mL, Sigma-Aldrich Chemie GmbH, München, Germany) (diluted in 500 mL/1,000 mL of 0.9% saline for the ponies/horses) was infused at a dosage of 10 ng/kg BW over 30 min. The animals were monitored for 3 h, using a modified pain score described by *Bussieres et al. (2008)*, which grades 13 different parameters on a scale from 0 (physiologic) to 3 (pathologic). Examined parameters were for example the rectal temperature, appetite and abdominal discomfort. Fourteen hours after the LPS infusion, blood samples were collected in tubes containing coagulation activator (Monovette; Sarstedt AG, Nuembrecht, Germany) for SAA determination. The tubes were centrifuged at $865 \times g$ for 10 min after 30 min of clotting time. Serum was harvested and gradually frozen from −20 to −80 °C.

## Liver tissue sampling

The animals were sedated with 0.04 mg/kg BW romifidine (Sedivet®; Boehringer Ingelheim Pharma GmbH & Co. KG, Ingelheim am Rhein, Germany) and 0.03 mg/kg BW butorphanol (Alvegesic®; CP-Pharma Handelsgesellschaft GmbH, Burgdorf, Germany) 15 h after the LPS challenge. Diazepam at a dose of 0.08 mg/kg BW (Diazepam-Lipuro®; Laboratoire TVM, Lempdes, France) and three mg/kg BW ketamine (Ursotamin®; Serumwerk Bernburg AG, Bernburg, Germany) were administered to induce general anesthesia. Inhalation anesthesia was maintained with isoflurane (CP-Pharma Handelsgesellschaft GmbH, Burgdorf, Germany). A 20-cm ventral midline incision was performed cranial to the umbilicus after aseptic preparation. Liver tissue (~two g) was collected using biopsy forceps. Additionally, adipose tissue was collected from several locations for another part of the study. One part of each tissue biopsy specimen was immediately flash frozen in liquid nitrogen (−196 °C) and stored at −80 °C. The second part was stored in formalin. The animals were orally administered 0.55 mg/kg BW flunixin twice a day for 3 days after surgery (Flunidol®; CP-Pharma Handelsgesellschaft GmbH, Burgdorf, Germany).

## Determination of BW

Body weight was obtained weekly using an electronic scale system for large animals (scale system: Iconix FX 1, Texas Trading, scale precision: 0.5 kg).

## BCS and CNS

Body condition score (*Carroll & Huntington, 1988*) and CNS (*Carter et al., 2009a*) were assessed on a scale ranging from 0 to 5 points. BCS and CNS were graded weekly by two independent evaluators (CS and DB). A mean of these two evaluators was calculated.

## Analysis of blood samples

Plasma ACTH levels were analyzed by a commercial laboratory using a chemiluminescence immunoassay (IDEXX GmbH, Ludwigsburg, Germany).

Serum liver enzyme activities (alkaline phosphatase (ALP), glutamate dehydrogenase (GLDH), AST, and GGT) and serum BA, TG and NEFA concentrations were analyzed using an automated chemistry analyzer (Roche Cobas C311; Roche Diagnostic GmbH, Mannheim, Germany).

SAA levels were determined by turbidimetry (ABX Pentra 400 analyzer, ABX Horiba; Axonlab, Montpellier, France).

Plasma glucose concentrations were determined using the GOD/POD method.

Serum insulin levels were analyzed using an immunoradiometric assay (IRMA, 125I; Demeditec Diagnostics GmbH, Kiel, Germany).

## Histological staging of hepatic steatosis

Hematoxylin-eosin staining was routinely performed on all liver biopsies and analyzed by an experienced histologist (MG). Steatosis was graded as follows: <5% lipid content of liver parenchyma: 0; 5–33%: 1; >33–66%: 2; >66%: 3.

## Analysis of hepatic mRNA levels

RNA was isolated using a commercial kit (RNeasy Lipid Tissue Mini Kit and Qiacube; Qiagen, AMBION, Inc., Germantown, MD, USA) according to the manufacturer's protocol. The RNA concentration was measured with a spectrophotometer (NanoVue® Plus; Healthcare Biosciences AB, München, Germany). RNA quality was determined using an Agilent 2100 Bioanalyzer (Agilent Technologies, Santa Clara, CA, USA). Two micrograms of RNA were transcribed into cDNAs in a thermocycler (Engine Peltier Thermal Cycler; Bio-Rad Laboratories GmbH, München, Germany) using two master mixes: (1) random primers and dNTP and (2) SuperScript II RT, 5x First Strand Buffer, and 0.1 M DTT (Thermo Fisher Scientific Inc., Schwerte, Germany). The genes of interest were IL-1β, IL-6, TNFα, cluster of differentiation 68 (CD68), chemerin, nuclear factor-κB (NF-κB), lipoprotein lipase (LPL) and FABP1. The 18S rRNA, hypoxanthine phosphoribosyltransferase 1 (HPRT1) and ribosomal protein L32 (RPL32) were chosen as reference genes (see Table 2 for primer sequences). Reference gene mRNAs were not altered by BW gain (*Pfaffl et al., 2004*). An RNA-probe was used for 18S rRNA quantification. A standard Taqman program (7500 Real Time PCR System; Thermo Fisher Scientific Inc., Schwerte, Germany) was performed for qPCR, with minor modifications. Two master mixes were utilized: Power SYBR Green PCR Master Mix for the genes detected with primers and Taqman Universal PCR Master Mix for the 18S RNA (Thermo Fisher Scientific Inc., Schwerte, Germany). The genes of interest were normalized to the geometric means of the three reference genes 18S, HPRT1 and RPL32.

## Statistics

The data were analyzed using a statistical software program (STATISTICA, version 12, RRID:SCR_014213; StatSoft GmbH, Hamburg, Germany). The data were analyzed for

**Table 2 Primer sequences used to analyze the levels of the genes of interest and reference genes.**

| | Forward (5′-3′) | Reverse (3′-5′) |
|---|---|---|
| IL-6 *Ungru et al. (2012)* | CCACCTCAAATGGACCACTACTC | TTTTCAGGGCAGAGATTTTGC |
| TNFα *Figueiredo et al. (2009)* | AAAGGACATCATGAGCACTGAAAG | GGGCCCCCTGCCTTCT |
| CD68 *Ungru et al. (2012)* | CTTTGGGCCAAGTTTCTCTTGT | AAGAGGCCGAGGAGGATCAG |
| HPRT1 *Bogaert et al. (2006)* | GGCAAAACAATGCAAACCTT | CAAGGGCATATCCTACGACAA |
| RPL32 *Bogaert et al. (2006)* | AGCCATCTACTCGGCGTCA | TCCAATGCCTCTGGGTTTC |
| IL-1β[#] | CGGCAATGAGAATGACCTGT | GCTTCTCCACAGCCACAATG |
| LPL[#] | ATTGTGGTGGACTGGCTGT | GCTCCAAGGCTGTATCCCAA |
| FABP1[#] | CAAGATCACCATCACCACAGG | GTCACAGACTTGATGCCTTTGA |
| Chemerin[#] | CATGGGAGGAAGCGGAAATG | CAGCTGAGCCTGTGTCTCTA |
| NF-κB[#] | GCTTTGTGACAAGGTGCAGA | ACGATCATCTGTGTCTGGCA |

**Notes:**
Five qPCR primers were newly designed and five primers were obtained from published data.
[#] Designed using http://primer3.ut.ee/. The specific equine cDNA sequences were provided by http://www.ensembl.org/index.html and the generated primers were validated in http://eu.idtdna.com/calc/analyzer to confirm the absence of hairpins, homodimers and heterodimers. The designed primers were created with two different modifications for each gene of interest and the more suitable primer was selected in preliminary tests. Primers were synthesized by biomers.net GmbH.

normal distributions using the Shapiro–Wilks test. ANOVA with repeated measurements was performed to analyze plasma glucose, serum insulin, serum TG and serum NEFA concentrations. Fisher's Least Significant Difference test was performed to identify significant differences. BCS, CNS, serum liver enzyme activities, serum BA concentrations, liver lipid content and mRNA levels of genes of interest were analyzed using nonparametric tests. Friedman's ANOVA was used to analyze the effect of time. When significant differences were observed, the Wilcoxon signed rank test with Bonferroni's correction was performed as post hoc test. The effects of the breed on nonparametric data were analyzed using the Mann–Whitney $U$ test. Correlations among variables were examined by calculating Spearman's correlation coefficients. Pain score values of each parameter at LPS challenge were added and described descriptively. Statistical significance was set to $P < 0.05$.

## RESULTS

Prior to the initiation of excess energy intake, all animals were assessed as metabolically healthy, according to the ACTH concentrations and results of the CGIT.

The 2 years of excess energy intake caused a significant increase in mean BW (± SD) of 29.9 ± 19.4% for ponies ($P = 0.0002$) and 17 ± 6.74% ($P = 0.00004$) for horses. Throughout the study, no significant differences in BW gain (%) could be found between the breeds. One pony developed an episode of laminitis during the second year of excess energy intake. Therefore, final samples were collected before the end of the study (July 2017) after complete recovery of clinical signs (pounding digital pulse, lameness). The pony received opioids once for pain relief 7 days before data collection. Laminitis occurred additionally in one horse at the end of the second year of excess energy intake. This horse received non-steroidal anti-inflammatory drugs for pain relief 20 days before sample collection. Time point of sampling was in accordance with the study design in

**Table 3 BCS and CNS in ponies and horses during 2 years of excess energy intake (data are presented as medians and 25th/75th percentiles).**

| Breed | Score | t0 | t2 | t5 |
|-------|-------|-----|-----|-----|
| Ponies | BCS | 2.3 (1.2/3.4)[a] | 3.6 (3.4/3.7)[a] | 3.9 (3.7/4.2)[b] |
|  | CNS | 2.5 (0.8/3)[a] | 2.8 (2.5/3.0)[a] | 3.5 (3.3/4.0)[b] |
| Horses | BCS | 2.7 (2.1/3.2)[ab] | 3.6 (3.5/3.6)[b] | 3.8 (3.7/3.9)[c] |
|  | CNS | 2 (1.8/2.3)[a] | 2.8 (2.8/3.0)[b] | 3.5 (3.5/4.0)[c] |

Note:
Different superscript letters indicate significant differences within a row.

**Table 4 Plasma glucose (mmol/L), serum insulin (µU/mL), serum NEFA (µmol/L) and serum TG (mmol/L) concentrations recorded in ponies and horses during 2 years of excess energy intake (data are presented as means ± SD).**

| Parameter | t0 | | t2 | | t5 | |
|-----------|------|------|------|------|------|------|
|  | Ponies | Horses | Ponies | Horses | Ponies | Horses |
| Glucose (mmol/L) | 3.53 ± 0.64[a] | 4.08 ± 0.21[b] | 3.93 ± 0.38[ab] | 4.52 ± 0.23[b] | 4.34 ± 0.86[b] | 4.41 ± 0.48[b] |
| Insulin (µU/mL) | 4.26 ± 1.36[ac] | 6.32 ± 2.35[c] | 7.93 ± 5.75[abc] | 9.3 ± 3.18[abc] | 13.9 ± 14.9[b] | 15.1 ± 10.3[b] |
| NEFA (µmol/L) | 119 ± 117[a] | 337 ± 381[b] | 208 ± 168[ab] | 211 ± 89[ab] | 352 ± 141[b] | 247 ± 87[ab] |
| TG (mmol/L) | 0.49 ± 0.19[a] | 0.27 ± 0.09[bc] | 0.41 ± 0.31[ab] | 0.24 ± 0.05[c] | 0.42 ± 0.2[ab] | 0.31 ± 0.08[bc] |

Note:
Different superscript letters indicate significant differences within a row.

the laminitic horse. No clinical signs of laminitis were present at this point. One pony developed hyperlipemia (serum TG: 14.4 mmol/L) at the end of the second year of excess energy intake beside good appetite. The pony was carefully monitored (e.g., appetite, behavior, lameness) and the pony recovered without medication within 14 days. Samples were collected after serum TG concentrations returned to the baseline value. The animals did not suffer from additional health problems due to excess nutrition such as colic. BCS and CNS increased significantly in ponies and horses during the 2 years of excess energy intake (Table 3). No significant differences in BCS and CNS were observed between the breeds. The CNS of the laminitic pony (4.5 points) was greater than the median of the respective cohort at t5, whereas the CNS of the laminitic horse (3 points) remained within the median of the respective cohort. The BCS and CNS of the lipemic pony were within the median of the cohort of the ponies (BCS: 3.75; CNS: 4).

Plasma glucose and serum NEFA concentrations were significantly lower in ponies compared to horses at t0. With BW gain plasma glucose and serum NEFA concentrations increased in ponies, but not in horses. No significant differences between the breeds were observed at t2 and t5 concerning these parameters (Table 4). Basal serum insulin concentrations increased significantly in both breeds from t0 to t5. The ponies showed significantly higher serum TG concentrations than horses at t0 and t2 (Table 4). Mean SAA concentrations in ponies and horses were below the reference range of 2.7 µg/mL at all three data collection points.

The sum of pain score points increased subsequently to the LPS infusion in the animals from basal mean values (t0: 1.4; t2: 0.5; t5: 1.5) to mean maximum values (t0: 6.1; t2: 6.7;

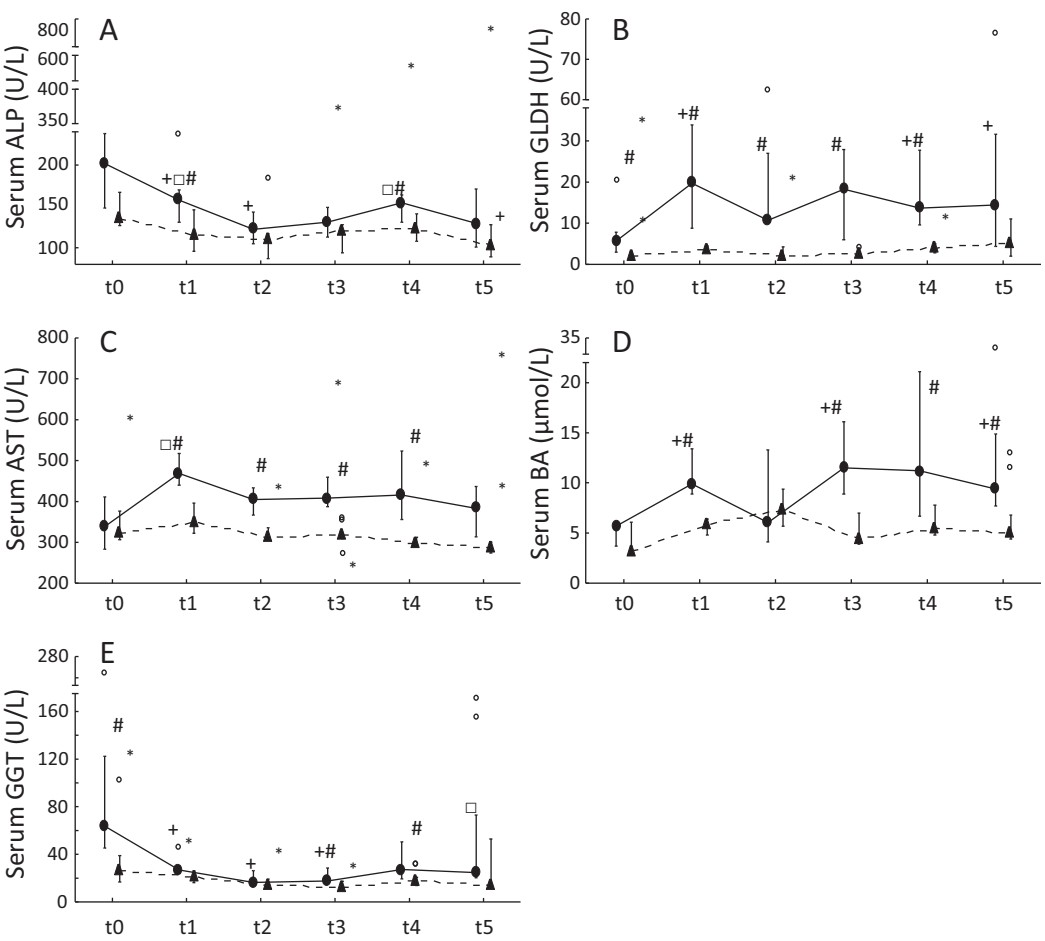

**Figure 1 Serum liver enzyme activities and serum BA concentrations in ponies and horses during 2 years of excess energy intake.** Serum ALP activities (A), serum GLDH activities (B), serum AST activities (C), serum bile acids (D) and serum GGT activities (E) in ponies ($N = 10$) (filled circles) and horses ($N = 9$) (triangles) at t0, t1, t2, t3, t4 and t5 (reported as medians (filled circles or triangles), 25th/75th percentiles (whiskers), outliers (blank circles) and extreme values (asterisk)); + significantly different from t0; □ significantly different from t2; # significantly different between ponies and horses at the certain time point.

t5: 6.1). Mean maximum values were reached 60 min after the LPS infusion at t0 and t2 and 30 min after the LPS infusion at t5.

## Serum liver enzyme activities

Serum ALP and GGT activities were significantly decreased in ponies after 1 year of excess energy intake. In ponies, serum GLDH activities and BA concentrations increased significantly from t0 to t5. Furthermore, significantly higher serum GLDH activities (t0: $P = 0.04$; t1: $P = 0.00002$; t2: $P = 0.02$; t3: $P = 0.00002$; t4: $P = 0.001$), serum AST activities (t1: $P = 0.0007$; t2: $P = 0.003$; t3: $P = 0.001$; t4: $P = 0.02$) and BA concentrations (t1: $P = 0.0004$; t3: $P = 0.001$; t4: $P = 0.03$; t5: $P = 0.03$) were observed in ponies compared to horses. The horses showed no significant increases in serum liver enzyme activities but exhibited a significant decrease in serum ALP activity from t0 to t5 (Fig. 1). Serum ALP

**Table 5 Staging of hepatic steatosis in ponies and horses during 2 years of excess energy intake (data are presented as numbers and as percentages of the breed).**

| Steatosis stage | t0 | | t5 | |
|---|---|---|---|---|
| | Ponies | Horses | Ponies | Horses |
| 0 | 8/10 (80%) | 6/9 (66.6%) | 6/10 (60%) | 8/9 (88.9%) |
| 1 | 2/10 (20%) | 3/9 (33.3%) | 2/10 (20%) | 1/9 (11.1%) |
| 2 | 0/10 (0%) | 0/9 (0%) | 1/10 (10%) | 0/9 (0%) |
| 3 | 0/10 (0%) | 0/9 (0%) | 1/10 (10%) | 0/9 (0%) |

activity, BA concentrations and GGT activity increased in the lipemic pony during the 2 years of excess energy intake (ALP: 6.2-fold increase; BA: sixfold increase; GGT: 2.3-fold increase). Serum liver enzyme activities and BA concentrations in the laminitic equines remained within the reference ranges, (*Köller, Gieseler & Schusser, 2014*) except for serum GGT activity at t5 in the laminitic pony (GGT = 73.1 U/L).

## Histological staging of steatosis

At t5 the percentage of individuals exhibiting a hepatic lipid content of more than 5% increased in ponies and decreased in horses. However, the majority of ponies and horses showed a constant steatosis grade 0 (Table 5). The pony suffering from hyperlipemia showed stage 3 steatosis, with more than 66% lipid-loaded hepatocytes at t5 (steatosis stage at t0: 0). Both laminitic equines steatosis stage 0 at t0. The laminitic horse stayed at steatosis stage 0 throughout the study, but the laminitic pony showed steatosis stage 1 at t5.

## Hepatic mRNA levels of genes of interest

Levels of the TNFα, IL-6, FABP1 and CD68 mRNAs were not significantly altered in ponies and horses throughout the observation period. Hepatic levels of the chemerin mRNA remained constant from t0 to t2 in both breeds. In ponies, the hepatic level of the chemerin mRNA increased significantly from t2 to t5. The horses showed a significant increase of the level of the chemerin mRNA from t0 to t5 and from t2 to t5 (Fig. 2). Levels of the NF-κB mRNA decreased significantly in horses from t2 to t5 ($P = 0.02$) and remained unchanged in ponies throughout the study. Regarding the breed-specific differences, the ponies showed significantly higher hepatic levels of the LPL ($P = 0.005$), NF-κB ($P = 0.01$) and IL-1β ($P = 0.045$) mRNAs compared with the horses at t5. The lipemic pony showed a higher level of the LPL mRNA than the median of the pony cohort (t0: 3.1-fold higher; t2: 2-fold higher; t5: 2.5-fold higher) at all time points. Furthermore, the level of FABP1 mRNA was 2.5-fold higher at t5 compared to the median of the pony cohort. No further notable deviations in the levels of the genes of interest in the laminitic and lipemic equines from the median values of the cohort were observed.

Significant correlations between the level of the LPL mRNA and serum BA, the hepatic lipid content and level of the CD68 mRNA were analyzed. Levels of the chemerin mRNA displayed significant correlations with the BCS, CNS and level of the NF-κB mRNA (Table 6). In ponies, a negative correlation was identified between serum ALP activity and

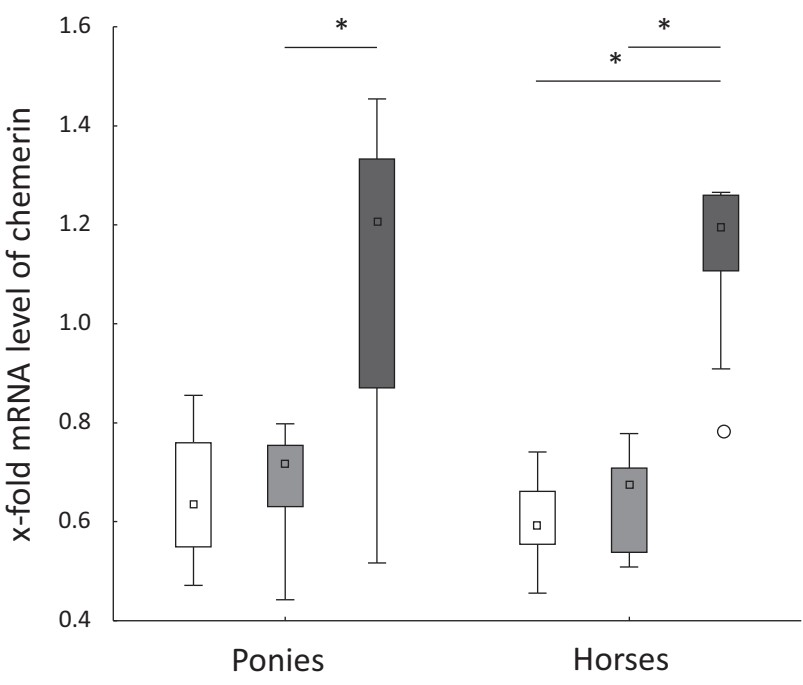

**Figure 2 Hepatic chemerin mRNA level in ponies and horses.** Fold changes in the hepatic levels of the chemerin mRNA at t0 (white), t2 (light gray) and t5 (dark gray) in ponies ($N = 10$) and horses ($N = 9$) (reported as medians (squares), 25th/75th percentiles (boxes), minimum and maximum values (whiskers), and outliers (circles)); significant differences are indicated by $^*$, no significant differences between ponies and horses were observed.               

**Table 6 Correlations between the level of the LPL and chemerin mRNAs with serum BA concentrations, hepatic lipid contents, BCS, CNS and the levels of the CD68 and NF-κB mRNAs.**

| Variables | Ponies ($N = 10$; $n = 30$) | | Horses ($N = 9$; $n = 27$) | |
|---|---|---|---|---|
| | $r^1$ | P-value | $r^1$ | P-value |
| Chemerin × BCS | 0.6 | <0.001 | 0.5 | 0.005 |
| Chemerin × CNS | 0.6 | <0.001 | 0.6 | 0.001 |
| Chemerin × NF-κB | −0.2 | 0.2 | −0.6 | 0.002 |
| LPL × BA | 0.4 | 0.02 | −0.3 | 0.2 |
| LPL × CD68 | 0.4 | 0.02 | 0.6 | <0.001 |
| LPL × hepatic lipid content | 0.4 | 0.02 | 0.01 | 0.9 |

**Note:**
  $r^1$ = Spearman's correlation coefficient.

age ($r = -0.4$; $P = 0.01$). No significant correlation between serum ALP activity and age was detected in horses.

## DISCUSSION

Previous studies of equines have been conducted to evaluate the impact of obesity on metabolic characteristics and inflammatory markers in adipose tissue (*Vick et al., 2007*; *Burns et al., 2010*). To the best of the authors' knowledge, this study is the first to investigate obesity associated hepatic alterations in equines. The study design was further

applied to compare ponies and horses during a long-term feeding period of excess energy intake. However, in comparison with previous studies (*Carter et al., 2009b*; *Siegers et al., 2018*) our equines gained less BW during excess energy intake. In the studies of *Carter et al. (2009b)* and *Siegers et al. (2018)*, the energy intake was provided by approximately 60% concentrate and 40% roughage. In contrast, the energy intake in the present study was covered by 40% concentrate and 60% roughage. Therefore, the diet of the present study contained less starch and more fiber in comparison to the aforementioned studies. These varying diet compositions may have caused the differences in BW gain. We have used this type of diet (1) for welfare reasons during a long-term observation period and (2) its closer relation to practical feeding regimens.

In humans, steatosis is considered a hepatic manifestation of metS. A standardized procedure for assessment of liver fat content is important, as the hepatic lipid content may vary between different hepatic regions. In the present study, liver sampling was conducted at the same site at the three data collection time points by laparotomy. Using this method, a standardized follow-up of the same area of the liver was possible. But, sampling of liver tissue in vivo is always restricted to a small hepatic region. However, in cows, the liver fat content differed less than 2% between different liver lobes, suggesting a homogenous distribution of fat (*Gerspach et al., 2017*). In consequence, the used liver sampling procedure is likely to provide reliable insight in hepatic fat storing in obesity. According to our hypothesis, the percentage of ponies exhibiting a hepatic lipid content of more than 5% increased from 20% at t0 to 40% at the end of the study. But unexpected, the percentage of horses with hepatic lipid content of less than 5% increased from 67% at t0 to 89% at the end of the study. These results are in line with findings in NEFA concentrations, which increased during the study in ponies but not in horses. As 60–80% of liver stored lipid is derived by circulating free fatty acids, the differences of NEFA concentrations between ponies and horses might have contributed to breed related differences in steatosis stages. The results of serum NEFA concentrations are described in detail by *Blaue et al. (2019)*. Concluding, ponies seem to be more susceptible to hepatic lipid accumulations in early stages of obesity according to the histological evaluation. Furthermore, we speculated that horses might not develop a steatotic liver in association with early obesity in contrast to humans. Healthy humans showing a BW gain of 5–15% significantly increased the liver fat content by 2.5-fold within 4 weeks (*Kechagias et al., 2008*), highlighting the profound differences between equine and human metabolism. However, differences might be explained by the 3.5-fold higher relative daily intake of fat in humans (*Kechagias et al., 2008*) compared to our equines. Accordingly, *De Meijer et al. (2010)* showed that the dietary fat content itself, independent from caloric intake, was a detrimental factor in the development of steatosis in mice. Despite the differences between ponies and horses, the majority of our equines did not develop steatosis within 2 years of BW gain. It has been described in Shetland ponies that subcutaneous tissue expandability is limited, while the expansion of the retroperitoneally adipose tissue proceeds (*Siegers et al., 2018*). It is one possible explanation that the expandability limit of the subcutaneous adipose tissue was not reached in the present study and therefore, fat was not stored extraordinary in retroperitoneal and intraabdominal sites like the liver.
The individual pony which developed hyperlipemia in the present study increased steatosis from stage 0 at t0 to stage 3 at the end of the study. Interestingly, this pony did not show the most prominent increase in BW, BCS or CNS. However, the question of whether the liver was steatotic before the onset of hyperlipemia or whether the lipid mobilization caused the elevated hepatic lipid content remained open.

The levels of the LPL and FABP1 mRNAs in the liver were determined to elucidate the role of hepatic fat metabolism in obesity. *Pardina et al. (2009)* described a significant increase in hepatic LPL mRNA levels in obese humans compared to healthy controls. The authors postulated that these changes contributed to the hepatic accumulation of TG, which favors steatosis. Accordingly, we identified a positive correlation between the hepatic level of the LPL mRNA and the hepatic lipid content in ponies. At t5, ponies showed a 2.4-fold higher level of the LPL mRNA compared to horses. We speculated that ponies may develop an increased risk of steatosis in cases of continuing the long-term excess energy intake. Data of increasing steatosis grade in the liver of ponies supported this assumption. FABP1 is known to facilitate the intracellular trafficking of long-chain fatty acids (*Glatz, Van Der Vusse & Veerkamp, 1988*). The level of the FABP1 mRNA is increased in humans with steatosis, probably as a compensatory mechanism for increased fat influx (*Higuchi et al., 2011*). Therefore, our expectation that the level of the FABP1 mRNA in mostly nonsteatotic equines would be unaltered was confirmed. Interestingly, the lipemic pony showed a three-fold elevation in the level of the LPL mRNA at t0 compared to the median of the ponies at t0. We speculated that the pony was already predisposed to developing a steatotic liver, even in the lean body condition. In this pony, high LPL mRNA level was maintained until t5, with 2.5-fold higher levels than the median of the ponies at the end of the study. Additionally, the pony showed a 2.5-fold higher level of the FABP1 mRNA than the median of the ponies at the end of the study, probably due to metabolic demands caused by high fat influx in the liver.

Besides liver lipid content, liver enzyme activities represent useful markers for hepatic metabolism. Serum ALP and GGT activities, exhibited either a significant decrease or remained unchanged in the present study. Serum ALP activities showed a significant decrease during the study in ponies and horses. According to *Gehlen, May & Venner (2010)*, elevated serum ALP activities might be associated with increased bone turnover in young horses. Therefore, changes in ALP activities in ponies and horses might be explained by age-related effects. Serum GLDH activity has interesting properties as marker of liver diseases, as GLDH is a liver-specific enzyme that is mainly located in the mitochondria in the centrolobular hepatocytes (*Schmidt & Schmidt, 1988*). We observed a significant increase in serum GLDH activity in the ponies from t0 to t5, but not in the horses. Serum GLDH activities in the ponies were 2.4-fold higher than the upper reference range of 8.9 U/L (*Köller, Gieseler & Schusser, 2014*) at the end of the study. Similar to serum GLDH activities, the ponies showed a significant increase in serum BA concentrations from t0 to t5. However, most of the ponies showed serum BA concentrations that remained within the reference range at the end of the study. While BA have long been known to mediate nutrient absorption, BA have recently emerged as signaling molecules for lipid and glucose metabolism (*Ma & Patti, 2014*). Furthermore, plasma BA levels

were shown to exhibit positive correlations with insulin resistance and type 2 diabetes in humans (*Haeusler et al., 2013*). Therefore, BA are not only a sensitive marker of liver diseases but also an important marker of metS and NAFLD. As further noninvasive marker, serum AST activities are increased in patients with NAFLD (*Sookoian et al., 2016*). In the present study, ponies showed significant higher serum AST activities compared to horses at four time points. As changes of serum GLDH activity and serum BA during BW gain occurred in ponies but not in horses and serum AST activities were higher in ponies than in horses, it is speculated that the liver of ponies was more affected by early obesity compared to horses.

Serum GGT activity of the laminitic pony exceeded the reference range at the end of the study. Accordingly, *Chameroy et al. (2011)* observed elevated serum GGT activities in 64.3% of obese horses with a history of laminitis. In addition, the steatosis stage of the laminitic pony increased from stage 0 at t0 to stage 1 at t5. In contrast, the laminitic horse showed neither an increase in serum liver enzyme activities nor an increase in hepatic lipid content during the study.

We determined the hepatic mRNA levels of proinflammatory cytokines to investigate whether the liver contributed to low-grade inflammation concomitant to obesity (*Vick et al., 2007*). The most prominent change in hepatic mRNA levels as BW increased was found for chemerin. Chemerin has been identified as an adipokine in mouse, rat and human adipocytes (*Bozaoglu et al., 2007*; *Goralski et al., 2007*) and has a regulatory role in adipogenesis and adipocyte metabolism (*Goralski et al., 2007*). In addition to adipose tissue, chemerin is expressed in the liver as well (*Pohl et al., 2017*). Although substantial experimental evidence supports a proinflammatory role for chemerin (*Weigert et al., 2010*; *Chakaroun et al., 2012*; *Döcke et al., 2013*), other studies have suggested that chemerin might have anti-inflammatory properties (*Cash et al., 2008*; *Luangsay et al., 2009*). Consistent with these equivocal results, discrepancies exist regarding the association of chemerin and NAFLD. According to *Deng et al. (2013)*, rodents with NAFLD displayed decreased hepatic levels of the chemerin mRNA compared to control rodents without NAFLD. In contrast, the consumption of a high-fat diet increased hepatic mRNA levels of chemerin in mice compared to animals fed a standard diet (*Krautbauer et al., 2013*). Unfortunately, the authors did not provide information about the fat content of the diet. Additionally, the chemerin mRNA levels tended to be higher in the liver of humans with NAFLD (*Krautbauer et al., 2013*). These contradictory findings might be explained by the various pathways in which chemerin is involved. After secretion, chemerin is converted into proinflammatory or anti-inflammatory peptides by different proteases, as reviewed by *Yoshimura & Oppenheim (2008)*. To date, a study investigating chemerin has not been performed in equines. In the present study, the hepatic level of the chemerin mRNA increased significantly during 2 years of excess energy intake in ponies and horses. In contrast to the upregulation of chemerin, other proinflammatory factors, such as CD68, TNFα, IL-6 and IL-1β, were not different between lean and obese equines. Notably, we observed a significant negative correlation between hepatic levels of the chemerin and NF-κB mRNAs in horses but not in ponies. NF-κB is a well-known activator of the transcription of proinflammatory cytokines. This result highlights a

possible anti-inflammatory role for chemerin. In this context, *Pohl et al. (2017)* found a downregulation of the chemerin mRNA levels in livers of humans suffering from a progressive form of NAFLD compared to humans suffering from steatosis alone. Consequently, chemerin represents a potentially interesting marker for obesity associated hepatic alterations and should be the focus of future studies in equines. A limitation of this study was that changes in chemerin levels were not verified at the protein level.

## CONCLUSION

We detected significant differences in parameters such as serum GLDH and AST activities, serum BA concentrations and levels of the LPL mRNA between ponies and horses. According to our hypothesis, these differences suggested that ponies may show a more pronounced dysregulation of hepatic metabolism in reaction to the early stages of obesity compared to horses. However, in contrast to our hypothesis, liver steatosis seemed not to be an integral part of the early stages of obesity, especially in horses, and may occur in ongoing equine obesity. Liver mRNA levels of well-established proinflammatory cytokines such as TNFα or IL-6 were not significantly upregulated in response to increasing BW. However, chemerin was identified as a potentially novel marker of the hepatic changes associated with obesity in equines. A longer period of BW gain or a higher degree of obesity might be necessary to obtain more significant findings for inflammation and steatosis in the liver.

## ACKNOWLEDGEMENTS

The authors are grateful to S. Berthold, D. Kern, A. Ruhland, S. Klemann and J. Tietke for providing technical support.

### Funding

This study was funded by the German Research Foundation (VE 225/9-1) and the University of Leipzig within the program of Open Access Publishing. There was no additional external funding received for this study. The funders had no role in study design, data collection and analysis, decision to publish, or preparation of the manuscript.

### Grant Disclosures

The following grant information was disclosed by the authors:
German Research Foundation: VE 225/9-1.
University of Leipzig within the program of Open Access Publishing.

### Competing Interests

The authors declare that they have no competing interests.

### Author Contributions

- Carola Schedlbauer performed the experiments, analyzed the data, prepared figures and/or tables, authored or reviewed drafts of the paper, approved the final draft.

- Dominique Blaue performed the experiments, authored or reviewed drafts of the paper, approved the final draft.
- Martin Gericke conceived and designed the experiments, performed the experiments, contributed reagents/materials/analysis tools, authored or reviewed drafts of the paper, approved the final draft.
- Matthias Blüher conceived and designed the experiments, performed the experiments, contributed reagents/materials/analysis tools, authored or reviewed drafts of the paper, approved the final draft.
- Janine Starzonek performed the experiments, authored or reviewed drafts of the paper, approved the final draft.
- Claudia Gittel conceived and designed the experiments, performed the experiments, authored or reviewed drafts of the paper, approved the final draft.
- Walter Brehm conceived and designed the experiments, performed the experiments, authored or reviewed drafts of the paper, approved the final draft.
- Ingrid Vervuert conceived and designed the experiments, performed the experiments, analyzed the data, authored or reviewed drafts of the paper, approved the final draft.

### Animal Ethics

The following information was supplied relating to ethical approvals (i.e., approving body and any reference numbers):

The Ethics Committee for Animal Rights Protection of the Leipzig District Government (No. TVV 32/15) approved the project in accordance with German legislation for animal rights and welfare.

### Data Availability

The raw data is available in the Supplemental Files.

### Supplemental Information

Supplemental information for this article can be found online at http://dx.doi.org/10.7717/peerj.7069#supplemental-information.

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
