# Peer review of "Impact of body weight gain on hepatic metabolism and hepatic inflammatory cytokines in comparison of Shetland pony geldings and Warmblood horse geldings"

_PeerJ, doi:10.7717/peerj.7069_

## Round 0.1 · original submission · Minor Revisions

The study is well presented, and reflected the novelty. I feel that the manuscript is dealing with a good topic but lacks in the quality of preparation. The main problem found in the manuscript is related to the some aspects of the methodology and technical corrections. Please describe the subject of age differences clearly in MS and inform why these parameters (SAA, plasma glucose, serum insulin, TG and NEFA concentrations) are not measured for winter months. It is necessary to improve the manuscript by examining the questions that need to be clarified in a way. Please be aware of the manuscript should be presented according to guidelines for authors of PeerJ. For your guidance, you can check the reviewers' comments. Thank you for giving us the opportunity to consider your work.

·

Basic reporting

The manuscript is well written in all its parts and literature is well covered and referenced.

Experimental design

The topic of the manuscript is appropriate for the Journal and the information is of significant interest to the Journal\'s readers.
Some clarifications that should be added to the material and methods should be added to the manuscript as described in the general comments for the author

Validity of the findings

The data is well presented and reflected the novelty
The conclusions are appropriately stated and connected with the research question.

Additional comments

The title accurately reflects the major findings of the work, however, I suggest to add “geldings to Shetland ponies and Warmblood horses.
It is useful for the authors to mention the reason for the selection of only geldings animals in their study.
Line 92-93: have age differences between the ponies and the horses been considered when comparing the results between two groups?

Line 119-121: Serum amyloid A (SAA), plasma glucose, serum insulin, serum triglycerides (TG) and serum non-esterified fatty acids (NEFA) concentrations were analyzed only in t0 (October 2015), t2 (October 2016) and t5 (December 2017). Have these parameters been considered in winter only?

Line 114: Did the authors used a 14-gauge-catheter for blood sampling from ponies?

Line 155-169: Could the authors clarify why they used surgical method rather than classical liver biopsy?
Could the surgical operation influnce animal appetite, weight gain and the results of studied blood parameters?

Line 231: Did the animals suffer from additional health problems due to excess nutrition such as colic?
Were there any observations regarding the consumption of drinking water during the study? kindly focus on these observations.
Line 237-238: Could the administration of non-steroidal anti-inflammatory drugs 20 days before sample collection in one animal have any impact on studied liver parameters?

Line 417-418: When discussing the results it is important to consider the time of sample collection (t0,2, 5). Also the impact of season on the results should be discussed.

·

Basic reporting

no comment

Experimental design

line 109: please explain basal state more clearly (e.g. basal serum levels etc.)
line 125-143: please explain more clearly at what timepoint which blood parameter was measured

Validity of the findings

Results: Pain scores from the LPS challenge are not given in the text or tables
line 252: basal insulin? levels during CGIT??

Additional comments

No comments, nice article with interesting results and discussion

---

## Round 0.2 · accepted · Accept

Thank you for responding to all comments and for revising the manuscript. Best wishes,

·

Basic reporting

The topic of the manuscript is appropriate for the Journal. Clear, unambiguous, professional English language used throughout.

Experimental design

Materials and methods section don't needs any changes. Literature well referenced & relevant.

Validity of the findings

No comments

Additional comments

No comments

·

Basic reporting

no comment

Experimental design

good improvements

Validity of the findings

no comment

Additional comments

nice improvements, no further remarks to the paper